# Comparison of Anti-Viral Activity of Frog Skin Anti-Microbial Peptides Temporin-Sha and [K^3^]SHa to LL-37 and Temporin-Tb against Herpes Simplex Virus Type 1

**DOI:** 10.3390/v11010077

**Published:** 2019-01-18

**Authors:** Maëva Roy, Lucie Lebeau, Céline Chessa, Alexia Damour, Ali Ladram, Bruno Oury, David Boutolleau, Charles Bodet, Nicolas Lévêque

**Affiliations:** 1Laboratoire Inflammation, Tissus Epithéliaux et Cytokines, LITEC EA 4331, Université de Poitiers, 86000 Poitiers, France; maeva.roy01@etu.univ-poitiers.fr (M.R.); celine.chessa@chu-poitiers.fr (C.C.); alexia.damour@univ-poitiers.fr (A.D.); charles.bodet@univ-poitiers.fr (C.B.); 2Laboratoire de Virologie et Mycobactériologie, CHU de Poitiers, 86000 Poitiers, France; lucie.lebeau79@orange.fr; 3Sorbonne Université, CNRS, Institut de Biologie Paris-Seine, IBPS, BIOSIPE, 75252 Paris, France; ali.ladram@sorbonne-universite.fr; 4Institut de Recherche pour le Développement (IRD), UMR 224 IRD-CNRS-Univ Montpellier 1 et 2 Maladies infectieuses et Vecteurs: écologie, génétique, évolution et contrôle (MiVegec), 34394 Montpellier, France; bruno.oury@ird.fr; 5IRD, UMR 177 IRD-CIRAD, Interactions Hôtes-Vecteurs-Parasites-Environnement dans les maladies tropicales négligées dues aux Trypanosomatidae (InterTryp), 34394 Montpellier, France; 6Sorbonne Universités, Centre d’Immunologie et des Maladies Infectieuses (CIMI-Paris), INSERM U1135, Eq1, 75013 Paris, France; david.boutolleau@aphp.fr; 7AP-HP, Hôpitaux Universitaires La Pitié Salpêtrière-Charles Foix, Service de Virologie, Centre National de Référence Herpèsvirus, 75652 Paris, France

**Keywords:** temporin, SHa, [K^3^]SHa, Tb, LL-37, herpes simplex virus type 1, keratinocyte, cytotoxicity, anti-viral, immunomodulation

## Abstract

Temporins are anti-microbial peptides synthesized in the skin of frogs of the *Ranidae* family. The few studies to date that have examined their anti-viral properties have shown that they have potential as anti-viral therapies. In this work, we evaluated the anti-herpes simplex virus type 1 (HSV-1) activity of the temporin-SHa (SHa) and its synthetic analog [K^3^]SHa. Human cathelicidin LL-37 and temporin-Tb (Tb), previously demonstrated to have anti-HSV-1 properties, were used as positive controls. We observed that SHa and [K^3^]SHa significantly inhibit HSV-1 replication in human primary keratinocytes when used at micromolar concentrations. This anti-viral activity was equivalent to that of Tb, but lower than that of LL-37. Transcriptomic analyses revealed that SHa did not act through the modulation of the cell innate immune response, but rather, displayed virucidal properties by reducing infectious titer of HSV-1 in suspension. In contrast, pre-incubation of the virus with LL-37 suggests that this peptide does not act directly on the viral particle at non-cytotoxic concentrations tested. The anti-HSV-1 activity of LL-37 appears to be due to the potentiation of cellular anti-viral defenses through the induction of interferon stimulated gene expression in infected primary keratinocytes. This study demonstrated that SHa and [K^3^]SHa, in addition to their previously reported antibacterial and antiparasitic activities, are direct-acting anti-HSV-1 peptides. Importantly, this study extends the little studied anti-viral attributes of frog temporins and offers perspectives for the development of new anti-HSV-1 therapies.

## 1. Introduction

Herpes simplex virus type 1 (HSV-1) is in the *Herpesviridae* family, within the *Alphaherpesvirinae* sub-family, a classification that also includes HSV-2 and varicella-zoster virus [1,2]. This virus is transmitted by oropharyngeal secretions from an infected individual to a susceptible individual, primarily during childhood. Transmission may also occur during sex as HSV-1 is increasingly detected at the genital site [3]. Following primary infection, the virus establishes a life-long latent infection in the host with subsequent re-activation episodes. Primary and recurrent infections can result in painful skin vesicles or mucosal ulcers that may require anti-viral treatment in severe or recurrent forms [3]. In cutaneous as well as in mucosal lesions, keratinocytes are the main target cells of the virus [4].

Current HSV-1 therapies are virostatics, consisting of nucleoside (acyclovir) or pyrophosphate (foscarnet) analogs [5]. These compounds inhibit virus replication by targeting the viral DNA polymerase, but do not inactivate the already formed infectious viruses [5]. They can effectively treat mucocutaneous HSV-1 infections if they are given during prodromes, or early in the presentation of symptoms, but are of limited effect on already formed lesions [5]. Foscarnet, however, has been shown to exhibit significant nephrotoxicity, making it less acceptable as a treatment [5]. In addition, anti-viral resistance to anti-HSV-1 drugs has been observed in long-term treated or immune-compromised patients [5]. Further drug development with virucidal activity, low toxicity, and low selection of resistant mutants, therefore, remains a priority objective.

Anti-microbial peptides (AMPs) are small peptides, from 12 to 50 amino acids in length, with a large structural diversity, including α-helix peptides, cysteine-rich β-pleated sheets forming peptides, as well as non-structured peptides containing a high percentage of one specific type of amino-acid. More than 1300 AMPs have been discovered in a large number of vertebrate, invertebrate, and plant species [6,7]. AMPs display anti-microbial activities against a wide range of bacteria, fungi, and parasites by permeating and destroying plasma membranes [8]. The rather non-intuitive use against intra-cellular pathogens explains the more limited number of studies assessing their anti-viral activities. Recent work, however, has shown that AMPs can act against intra-cellular forms of parasites or structures internal to the parasite itself [9]. Against viruses, activity of AMPs can be both direct, by altering the virus particle or inhibiting its replication cycle, and indirect, by potentiating the host immune response caused by the infection [10].

An excellent example of an AMP, with demonstrated anti-HSV-1 activity, is the human cathelicidin LL-37, a 37-residue α-helical peptide that is synthesized by mast cells, neutrophils, natural killer, and epithelial cells, such as keratinocytes [11,12,13]. LL-37 and other amphipatic helix peptides act both directly on the viral envelope, by perforation, and on the cell membrane, by saturation of the attachment receptors of the virus, heparan sulfates, as well as indirectly by immune-modulation [14,15,16,17,18,19,20]. Recently, the amphibian anti-microbial peptide, temporin-Tb (Tb), has also been shown to inhibit HSV-1 infection in vitro [21]. AMPs, thus, represent a potentially promising field of research for new HSV-1 therapies.

Temporin-SHa (SHa) is an anti-microbial peptide of 13 residues originated from a precursor of the dermaseptin super-family and produced by cutaneous granular glands of the North African ranid frog *Pelophylax saharicus* from precursors of the dermaseptin superfamily [22]. It has, as do all temporins, a low net positive charge (+2) and is C-terminally amidated. Its amphipathic α-helical structure allows interaction with microbial cytoplasmic membrane, thereby promoting pore formation and membrane disruption [8,23]. [K^3^]SHa is a synthetic analog of SHa obtained by substituting the serine in position 3 with lysine [9]. The goal of this substitution was to increase the net positive charge of SHa to a value of +3, while retaining the α-helical structure, to enhance electro-static interaction of the peptide with the negatively charged membrane of the microorganisms, and thereby improve anti-microbial activity and reduce its cytotoxicity [9]. SHa and [K^3^]SHa have a broad-spectrum anti-microbial activity towards Gram-positive and Gram-negative bacteria, yeasts, and trypanosomatids (*Leishmania* and *Trypanosoma* parasites) at the micromolar concentration [9,22,23,24]. However, to date, their anti-viral properties have not been evaluated.

In this report, we evaluated the anti-HSV-1 properties of SHa and [K^3^]SHa. Their cytotoxicity against primary human keratinocyte cultures was initially determined, after which we compared their anti-viral activities with those of LL-37 and Tb during HSV-1 infection in the same cell culture system. Finally, we initiated the first step toward characterization of the mechanism of action of SHa on HSV-1 by pre-treatment of a viral inoculum before titration. Its indirect anti-viral properties were also assessed by transcriptomic analysis of the cell anti-viral response using LL-37 as a positive control.

## 2. Materials and Methods

### 2.1. Peptides

Synthetic human LL-37 was purchased from Invivogen (San Diego, California CA, USA). Temporin-SHa (SHa, molecular weight (MW) = 1380.76 g.mol^−1^) and its analog [K^3^]temporin-SHa ([K^3^]SHa) (MW = 1421.86 g.mol^−1^) were synthesized using solid-phase FastMoc chemistry on an Applied Biosystems 433A automated peptide synthesizer (Peptide Synthesis Platform, IBPS, Sorbonne University, Paris, France), then purified by semi-preparative reversed-phase high-performance liquid chromatography (RP-HPLC), as previously described [23]. The purity of the synthetic temporins was assessed by analytical RP-HPLC, using an Aeris PEPTIDE column (XB-C18, 3.6 µm, 4.6 × 250 mm, Phenomenex, Le Pecq, France) eluted at 0.75 mL/min by a 20–70% linear gradient of acetonitrile (1% ACN/min) in 0.1% trifluoroacetic acid/water, followed by MALDI-TOF mass spectrometry analysis (Mass Spectrometry and Proteomics Platform, IBPS, Sorbonne University, Paris, France). A 1 mM stock solution of SHa and [K^3^]SHa was prepared by dissolving lyophilized synthetic peptides in sterile MilliQ water. A stock solution of 1 mM temporin-Tb (MW = 1391.80 g.mol^−1^) was prepared in the same manner from synthetic peptides provided by Genscript (Piscataway, NJ, USA).

### 2.2. Virus and Virus Stock Production

The HSV-1 strain was isolated at the Poitiers University Hospital in 2017, following inoculation of the human lung fibroblast cell line MRC-5 (PD13 banked from NIBSC PD7, RD Biotech, Besançon, France) with samples of oral mucosal lesions. For virus production, monolayers of Vero cells (ATCC CCL-81, Manassas, VA, USA) in 150-cm^2^ tissue culture flasks were inoculated with HSV-1 at multiplicity of infection (MOI) of 0.01 TCID_50_ per cell. After 72 h at 37 °C, supernatants from infected cells were collected. Cellular debris was removed by centrifugation at 1000× *g* for 10 min and the cleared supernatant containing the virus was stored in presence of sucrose (30%) and N-[Hydroxyethyl] piperazine-N’-[2-ethanesulfonic acid] (HEPES) buffer (5%) at −80°C until use. The virus titer was measured by end-point dilution assay on Vero cells. Briefly, Vero cells were seeded at 4 × 10^3^ cells/well in 100 µL Dulbecco’s modified essential medium (DMEM, Gibco, Gaithersburg, MD, USA) supplemented with 2% de-complemented fetal bovine serum (FBS, Gibco, Gaithersburg, MD, USA ) in 96-well flat-bottom plates the day before titration. Viral suspensions were successively diluted from 10^−1^ to 10^−9^ in the same medium, after which 100 µL of each dilution were deposited in a row of 6 wells. The plates were observed daily and after 96 h of incubation at 37 °C in an atmosphere containing 5% CO_2_, the plates were read for titer. The wells in which the cells showed a cytopathic effect were considered positive for viral infection. The titer of the viral suspension was then determined using Kärber’s method for assessing the 50% tissue culture infective dose (TCID_50_) [25]. The titer of the virus stock preparation was 10^6.53^ TCID_50_/mL.

### 2.3. Isolation and Culture of Normal Human Epidermal Keratinocytes from Skin Samples

The use of all human skin samples for research studies was approved by the Ethics Committee (committee for the protection of persons) Ouest III (project identification code: DC-2014-2109). After the provision of fully informed consent, normal abdominal or breast skin was obtained from patients undergoing plastic surgery. Small pieces of skin were thoroughly rinsed with phosphate-buffered saline solution free of calcium and magnesium (PBS, Gibco, Gaithersburg, MD, USA) after removal of the fat. Skin was minced into fragments of about 125 mm^2^ using sterile scalpel blades. Skin samples were then incubated overnight at 4 °C in a 25 units/mL dispase II solution (Life Technologies, Carlsbad, CA, USA) to gently digest the tissue. Epidermal sheets were removed from the dermis using sterile pliers, then keratinocytes were dissociated by trypsin digestion (trypsin 1× in PBS) (trypsin-EDTA, Gibco, Gaithersburg, MD, USA) for 15 min at 37 °C. The cell suspension was then filtered through a 280 µm sterile filter. An equal volume of DMEM supplemented with 10% FBS was added, and the suspension was centrifuged at 300× g for 10 min at room temperature to pellet cells. Keratinocytes were re-suspended in Keratinocyte-Serum Free Medium (K-SFM, Invitrogen, Life Technologies, Carlsbad, CA, USA) supplemented with bovine pituitary extract (25 μg/mL, Invitrogen, Life Technologies) and recombinant epidermal growth factor (0.25 ng/mL, Invitrogen, Life Technologies). The cell concentration was determined using a Malassez cell, then 1 × 10^7^ cells were seeded into each 75-cm^2^ tissue culture flask in K-SFM supplemented with bovine pituitary extract and recombinant epidermal growth factor. The cultures were incubated at 37°C in a humidified atmosphere with 5% CO_2_ until approximatively 80% confluence was reached (usually between 1 to 3 weeks) after which the cultures were rinsed with PBS, trypsinized to release them from the plastic, washed and resuspended in complete DMEM medium, made 10% in DMSO (Sigma-Aldrich, Saint Louis, MO, USA) and stored frozen in liquid nitrogen until use. For use in virus inoculation studies, keratinocytes were seeded in sterile 24-well culture plates at a density of 2.5 × 10^5^ cells/well in 1 mL of K-SFM, supplemented with bovine pituitary extract and recombinant epidermal growth factor, and cultured to 80 % confluence (usually between 1 to 3 weeks). Cells were then starved overnight in un-supplemented K-SFM before infection. Human primary keratinocytes, that were used in each biological replicate were from different individuals.

### 2.4. Determination of Keratinocyte Viability

Primary keratinocytes were cultured in 96-well plates at 4 × 10^4^ cells per well in 0.1 mL K-SFM until approximately 80% confluence before being treated by various concentrations of AMPs for 24 h. Cell viability was assessed using the cell proliferation kit II (XTT) (Roche Diagnostics GmbH, Mannheim, Germany) according to the manufacturer’s protocol. The XTT labeling mixture was added after 24 h of incubation in the absence or presence of peptides at the indicated concentrations.

### 2.5. Anti-Viral Assays

The anti-viral properties of the AMPs studied were first assessed by evaluating their impact on growth kinetics of the virus inoculated on primary human keratinocytes. Keratinocytes were incubated with one of the four AMPs at progressively increasing non-cytotoxic concentrations for 1 h before addition of HSV-1 at an MOI of 0.1 TCID_50_ per cell. After 1 h of incubation with the virus and AMPs at 37 °C, the cell culture supernatant was replaced with fresh medium containing the peptides at the same concentrations used for pre-inoculation with virus. After 24 h, supernatant was removed and the cell monolayer was then lysed, using 2.4 mL of the NucliSENS^®^ Lysis Buffer (bioMerieux, Marcy-l’Etoile, France). The viral DNA concentrations in the lysates were determined by qPCR assay as described below.

The virucidal properties of LL-37 and temporin-SHa directly on the virus were assessed by pre-incubating 0.1 mL of the virus stock (described above) with peptides for 1 h at 37 °C before titration, by end-point dilution assay using Vero cells, as described above. The viral titer obtained was compared to that of the untreated virus suspension assayed in the same manner.

### 2.6. Nucleic Acid Extraction

The total nucleic acids from keratinocyte lysates were extracted on NucliSENS^®^ easyMAG^®^ automated system (bioMerieux, Marcy-l’Etoile, France) according to the manufacturer’s recommendations. The nucleic acid concentrations and purity were determined using the Nanodrop 2000 spectrophotometer (Thermo Fisher Scientific, Waltham, MA, USA).

### 2.7. Viral DNA Quantitation by qPCR

HSV-1 genome quantification in keratinocyte lysates was performed by qPCR. The set of forward (5’-CATCACCGACCCGGAGAGGAAC-3’) and reverse (5’-GGGCCAGGCGCTTGTTGGT TA-3’) primers were used to amplify a 92-bp fragment from the UL30 DNA polymerase gene, as previously described [26]. qPCR was performed in 96-well plates with a Light Cycler 480 system (Roche Diagnostics GmbH, Mannheim, Germany). A reaction mixture contained 5 µL of AceQ SYBR Green qPCR Master Mix (Vazyme Biotech, Nanjing, China), 1 μM forward and reverse primers, and 4 µL of nucleic acid extract in a total volume of 10 µL. PCR conditions were as follow: 2 min at 50 °C, 10 min at 95 °C, followed by 45 amplification cycles consisting of 15 s at 95 °C and 40 s at 60 °C. For viral DNA quantitation, a standard curve was generated in each real-time amplification from a serial 10-fold dilution of a positive control of a known HSV-1 DNA concentration. Each standard dilution was assayed in a duplicate. The viral DNA concentrations in the cell lysates were expressed as viral DNA copies per ng of extracted nucleic acids from infected keratinocytes.

### 2.8. Transcriptomic Analysis of the Innate Anti-viral Immune Response in Keratinocytes

To isolate RNA, the DNA in total nucleic acid extracts of infected keratinocytes was degraded using a Turbo DNA-free Kit (Invitrogen, Life Technologies, Carlsbad, CA, USA) according to the manufacturer’s instructions. RNA was reverse transcribed using SuperScript II kit (Invitrogen, Life Technologies, Carlsbad, CA, USA). Quantitative real time PCR was performed as described above using 1 μM forward and reverse primers (Table 1), designed using Primer 3 software (bioinfo.ut.ee/primer3-0.4.0/) and 12.5 ng of cDNA template in a total volume of 10 µL. PCR conditions were as follows: 5 min at 95 °C, 40 amplification cycles comprising 20 s at 95 °C, 15 s at 64 °C and 20 s at 72 °C. Samples were normalized with regard to two independent control housekeeping genes (Glyceraldehyde-Phospho-Dehydrogenase and 28S rRNA gene) and reported according to the ΔΔCT method as RNA fold increase: 2^ΔΔCT^ = 2^ΔCT sample − ΔCT reference^.

### 2.9. Statistical Analysis

Results were analyzed by GraphPad Prism, version 5 (GraphPad Software, La Jolla, CA, USA). The statistical significance of the difference between two groups was evaluated by the non-parametric Mann-Whitney test. Differences were considered to be significant at *p* < 0.05.

## 3. Results

### 3.1. Determination of Toxicity of Temporins and LL-37 on Human Primary Keratinocytes

We evaluated the cytotoxicity of the anti-microbial peptides to human primary keratinocytes (Figure 1). The cell cultures were incubated in the presence of increasing concentrations of peptides to measure cell viability after 24 h of exposure. LL-37 showed little or no significant impact on cell viability at concentrations of 1.25 and 2.5 μM but, at 5 µM, viability decreased to 80%. Based on previously published data, Tb was tested at concentrations ranging from 5 (7 µg/mL) to 60 µM (84 µg/mL) [21]. Significant cytotoxicity was only observed at the highest concentration used (60 μM) with a cell viability of 65%. Temporin-SHa was cytotoxic at 20 μM, showing a 77% cell viability, while [K^3^]SHa significantly decreased cell viability from 10 μM (74% cell viability) compared to untreated cells. The concentrations maintaining a cell viability greater than, or equal to, 90% were, therefore, used to test the anti-viral activity of the peptides. This ensured biasing of the results, that were obtained in viral replication, was minimized due to the significant cell mortality.

### 3.2. Assessment of HSV-1 Replication in Human Primary Keratinocytes Incubated with the Cathelicidin LL-37 and Temporins

Replication of the virus was assessed by assaying viral DNA levels in cells treated with the peptides. Human primary keratinocytes were inoculated for 24 h with 0.1 MOI of HSV-1, without (positive control) or with previously-determined, non-cytotoxic increasing concentrations of the AMPs used in the above experiment (Figure 1). The viral DNA concentration was subsequently measured in the cell monolayer by qPCR to assess the effect of the peptides on HSV-1 replication (Figure 2). The results showed that the viral DNA measured in the untreated cells reached 2.1 × 10^5^ copies per ng of total extracted nucleic acids at 24 h post-infection. The concentration of viral DNA was significantly reduced in cells treated with the two LL-37 concentrations of 1.25 and 2.5 µM causing a decrease of about one log (5.3 × 10^4^ (75% reduction) and 1.2 × 10^4^ copies (94%), respectively). Whereas, the other peptides were not as effective. SHa inhibited viral replication two-fold (52%) at the highest concentration tested, corresponding to 10 μM (1.0 × 10^5^ copies in treated cells). [K^3^]SHa significantly reduced the viral DNA concentration compared with non-treated cells from 2.5 µM. Viral DNA concentrations measured were 1.1 and 0.9 × 10^5^ copies at 2.5, and 5 µM, respectively, demonstrating a two-fold reduction (48% and 57%) that is equivalent to that obtained with SHa. Tb inhibited viral replication at 5 μM, but a significant reduction of HSV-1 replication was only observed at 40 μM, with a viral DNA concentration per ng of total cell nucleic acids equal to 0.8 × 10^5^ copies or equivalent to a reduction of less than 3 times (62%) compared with untreated cells. For all peptides tested, the observed pattern suggested a dose-dependent inhibition of viral replication.

### 3.3. Immunomodulatory Properties of LL-37 and SHa in HSV-1 Infected Keratinocytes

To initially explore the mechanism of anti-viral activity assessed during keratinocyte infection, we chose to study the effect of 2 out of the 4 peptides in the cell innate immune response: LL-37, already known for its immunomodulatory properties, and SHa, which has not previously been reported to have an impact on the innate response [14,17,18,19,27]. The expression profile of cellular anti-viral responses in the presence, and in the absence, of LL-37 and Sha, was studied by transcriptomic analyses carried out on the same samples as those used to measure inhibition of HSV-1 replication in keratinocytes treated with AMPs. In addition, the pro-inflammatory properties of LL-37 and SHa were evaluated by incubating keratinocytes with peptides alone. The mRNA expression levels of interferon (IFN) β, as well as five interferon stimulated genes (ISG) (IRF7, IFIT1, OAS1, ISG20 and viperin), which are known for their anti-viral activities, were quantified in cells inoculated with peptides or virus alone, as well as with virus in the presence of the strongest non-cytotoxic concentrations of LL-37 (2.5 µM) and SHa (10 µM) (Figure 3) [28,29,30,31]. In keratinocytes infected with the virus alone, we observed a strong induction of IFNβ expression with a 1360 mRNA fold increase in comparison with uninfected keratinocytes at 24 h post-infection (Figure 3A). The induction of ISG expression was much lower ranging from 1.2 for IRF7 (Figure 3B) to 4.5-fold increase for IFIT1 (Figure 3D). In cells infected with HSV-1 in the presence of LL-37, the mRNA levels of all the ISGs tested were increased despite a lower viral replication. A significant induction of ISG expression by comparison with both, uninfected cells and cells inoculated with the virus or the peptide alone, ranging from 6 fold for ISG20 (Figure 3E) to 36 fold for viperin (Figure 3F), was assessed. The expression of IRF7, a multi-functional transcription factor that can activate ISG expression, was significantly induced by LL-37 in infected keratinocytes (Figure 3B), a phenomenon which was not observed for IRF1 and IRF3 expression [32]. However, the IFNβ mRNA level was increased at a lower level than in infected, but untreated cells (Figure 3A). No induction of IFNα, another type I IFN, and type III IFN (IFN-λ1 and IFN-λ2) expression was observed in infected cells in the presence or absence of LL-37.

The treatment of keratinocytes, by SHa during HSV-1 infection, did not modify the expression profile of the components of the cellular anti-viral response studied compared to untreated infected cells (Figure 3). A weaker IFNβ mRNA level was found in infected cells in the presence of SHa, and this may be related to the reduced HSV-1 replication in keratinocytes treated with this peptide (Figure 3A).

Finally, at the tested concentrations, LL-37 and SHa alone did not exhibited significant pro-inflammatory properties on primary keratinocytes (Figure 3).

### 3.4. Virucidal Properties of LL-37 and SHa

The virucidal properties of LL-37 and SHa were assessed by measuring residual infectious titers, following pre-incubation of an HSV-1 suspension in the presence of a high concentrations of these two peptides (2.5 and 10 µM, respectively) for 1 h. The infectious titer measured in the presence of the peptides was compared to that of the untreated virus suspension (Figure 4). This experiment showed that treatment with LL-37 did not alter the infectious titer of the virus suspension, whereas incubation in the presence of SHa caused a significant loss in the infectious titer of more than 2 logs (2.8 × 10^7^ in the absence vs. 1.8 × 10^5^ TCID_50_/mL in the presence of SHa (99.4% reduction), suggesting a direct virucidal activity mechanism.

## 4. Discussion

The aim of this study was to evaluate the activity of SH-temporins against a pathogenic human herpes virus, HSV-1, an enveloped DNA virus frequently involved in cutaneo-mucous infections [3]. Based on the viral tropism, we tested the ability of SHa and [K^3^]SHa to inhibit the replication of HSV-1 during infections of primary human keratinocyte cultures [4]. We used human cathelicidin LL-37 and temporin-Tb, both of whose anti-HSV-1 activity had already been demonstrated, as reference peptides for the comparison of cytotoxicity, anti-viral properties, and mechanisms of action [20,21,33]. Tb also belongs to the temporin family, sharing the same structural characteristics as SHa and [K^3^]SHa [34]. Isolated from skin secretions of the European frog *Rana temporaria*, it has a sequence homology of 49.9% with SHa. Like SHa and [K^3^]SHa, Tb showed activity against both Gram-positive and Gram-negative bacteria, as well as against Candida albicans, the promastigotes, and amastigotes of *Leishmania* [34,35,36,37,38].

The non-cytotoxic concentrations, initially determined, were used to evaluate the anti-viral activities of each of the peptides. Of the four, LL-37 showed the highest anti-HSV-1 activity. Interestingly, in our model, this was not related to virucidal activity, since pre-incubation with LL-37 did not reduce the HSV-1 titer. The study of the ISG profiles then clearly showed the ability of LL-37 to potentiate the cellular innate immune response caused by HSV-1 infection, despite a reduced viral replication. The expression of multiple ISGs, including IFIT1, OAS1, ISG20 and viperin, with proven anti-viral or, specifically, anti-HSV-1 activity, was enhanced in the presence of LL-37 [28,29,30,31]. Moreover, as IFNβ expression was not stimulated in infected cells in the presence of LL-37, our results suggest that the induction of the anti-viral response by this peptide may occur via an IFNβ-independent manner. It was previously reported that ISG expression can be directly stimulated by some IRFs in an IFN-independent pathway and that HSV-1 can induce an interferon-independent ISG response [39,40,41]. We observed that the expression of IRF7, an ISG that can activate ISG expression in the absence of IFN signaling, was significantly induced in keratinocytes infected with LL-37 [32]. This suggest that the IRF7 pathway is involved in LL-37 induction of keratinocyte anti-viral response, during HSV-1 infection. The immunomodulatory properties of LL-37 have been established previously, and could explain its anti-HSV-1 activity instead of, or in addition to, the disruption of the viral envelope, which is its main anti-viral mechanism reported so far [42,43]. LL-37 was shown to increase IFNβ-1 mRNA expression induced by poly (I:C), a synthetic analog of viral double-stranded RNA, in human epidermal keratinocytes, leading to an enhanced anti-viral activity against HSV-1 [17,18,19]. It was also reported that the addition of LL-37 to rhinovirus-infected human bronchial epithelial cells enhanced IL-6 and CCL-2 production, and that LL-37-mediated production of chemokines induces recruitment of leukocytes to the site of infection, thereby contributing to the clearance of the viral infection [17,27]. In this study, we showed that LL-37 strengthened the anti-viral defenses of primary keratinocytes during HSV-1 infection.

In contrast, SHa did not possess immunomodulatory properties on the cell anti-viral response, since the expression of the ISGs, tested in the infected keratinocytes, was not modified in the presence of the peptide. Instead, its anti-viral mechanism of action appears to be mainly extra-cellular and virucidal. Indeed, the pre-incubation of the virus in the presence of SHa resulted in a significant loss in the infectious titer of the HSV-1 suspension that could suggest the direct inactivation of the virus by the peptide, perhaps through degradation of the viral envelope as previously shown for Tb [21]. This direct extra-cellular anti-viral activity was far superior to the reduction of the HSV-1 replication assessed at 24h post-infection of keratinocytes treated with SHa prior infection (2 logs vs. two-fold reduction). In the second assay, both the extra- and intra-cellular anti-viral effects of the peptide were investigated. However, a more limited extra-cellular exposure of the virus particle to the peptide, due to rapid intra-cellular virus penetration into keratinocyte following inoculation, could explain the lower anti-viral activity observed. However, an intra-cellular activity of SHa in addition to a probable detergent-like role cannot be definitively ruled out. Indeed, SHa and [K^3^]SHa have demonstrated an ability to act on intra-cellular forms of *Leishmania* by mitochondrial membrane depolarization or DNA fragmentation, leading to apoptotis-like death, in addition to primary membranolytic mechanism [9]. *In toto*, the intra-cellular anti-viral activity of the three temporins tested here are modest, especially compared to that of LL-37, but their direct, extra-cellular, virucidal effects offer perspectives for the development of new therapies. Engineered variations in their structure might help to enhance inhibition of virus replication while limiting the cytotoxic effects.

In summary, we have demonstrated that SHa and [K^3^]SHa have previously unsuspected anti-HSV-1 properties in addition to their reported antibacterial and antiparasitic activities. Our results are consistent with SHa acting primarily and directly on the viral particle rather than indirectly through an immunomodulatory mechanism. However, the exact nature of these agents’ anti-viral mechanism of action remains to be achieved. The ability of these peptides to act on the pre-fusion stages, such as the saturation of receptors or intra-cellular transport, as well as to inhibit intra-cellular steps of virus replication, such as viral transcription or translation, and virus assembly, will require further investigations. In addition, their activity against drug-resistant HSV-1 strains must be evaluated even if the main modes of action of temporins identified in this study, which do not target the viral DNA polymerase and do not require phosphorylation by viral thymidine kinase to be active, make the existence of cross-resistance with current anti-HSV-1 therapies unlikely. Finally, the ability of HSV-1 to develop resistance to temporins-SH will need to be tested.

Overall, and because of their diversity, AMPs offer a wide field of investigation for the discovery of new anti-viral drugs with virucidal activity. For example, the genome-wide analysis of the anti-microbial peptides, that are isolated from the snake *Python bivittatus*, recently identified new cathelicidins. These could have an anti-viral activity equivalent to that of LL-37, but with less cytotoxicity [44]. Nevertheless, high-throughput, sensitive screening methods for evaluating, both their cytotoxicity to human cells and their anti-viral properties against naked and enveloped viruses, need to be further developed. Finally, their use in therapy must first also solve the problem of route of administration. If the systemic pathway is difficult to consider because of a rapid degradation of AMPs, as for any peptide, their use in the local application of topical treatment directly on the cutaneous lesions may be more relevant. Seen in this light, our results suggest that a combination of one temporin, for their virucidal pre-fusion activity, and LL-37, for its immunomodulatory properties, would be particularly relevant for obtaining maximum anti-viral effect.

## Figures and Tables

**Figure 1 viruses-11-00077-f001:**
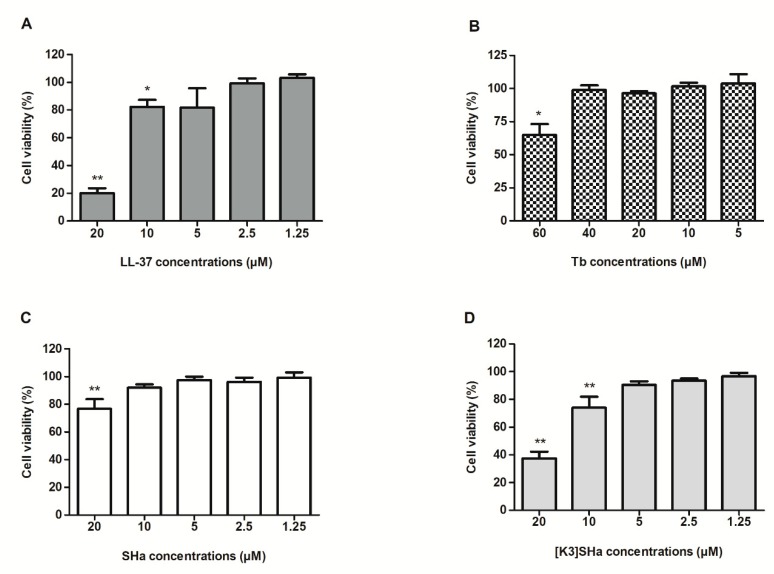
Cytotoxicity of LL-37 and temporins to human primary keratinocytes. Cell proliferation kit II (XTT) assays were performed to assess human primary keratinocytes viability after 24 h of treatment with LL-37 (A), temporin-Tb (Tb) (B), SHa (C) and [K^3^]SHa (D). Data are represented as mean + standard error of mean (SEM) of at least four independent experiments. **p* < 0.05 and ***p* < 0.01 compared with the untreated control.

**Figure 2 viruses-11-00077-f002:**
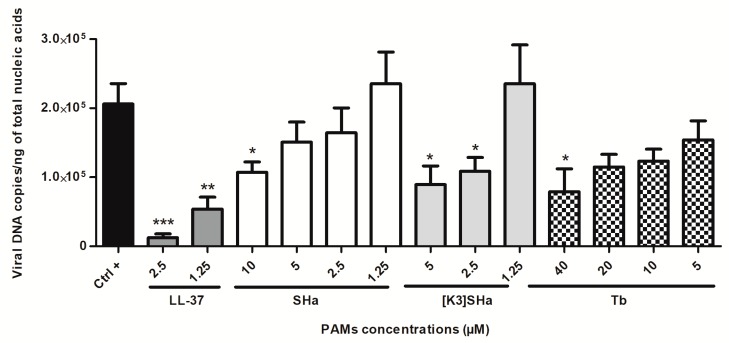
Herpes simplex virus type 1 (HSV-1) replication in human primary keratinocytes treated with LL-37, SHa, [K^3^]SHa and Tb. Primary keratinocytes were treated with the four AMPs at various concentrations for 1 h before being infected with HSV-1 at an multiplicity of infection (MOI) of 0.1 for 24 h. Viral DNA concentrations were determined in cell lysates by qPCR assay and expressed as viral DNA copies per ng of total cellular nucleic acids extracted from infected keratinocytes. Data are represented as mean + SEM of at least four independent experiments. **p* < 0.05, ***p* < 0.01 and ****p* < 0.01 compared with the infected control without AMPs.

**Figure 3 viruses-11-00077-f003:**
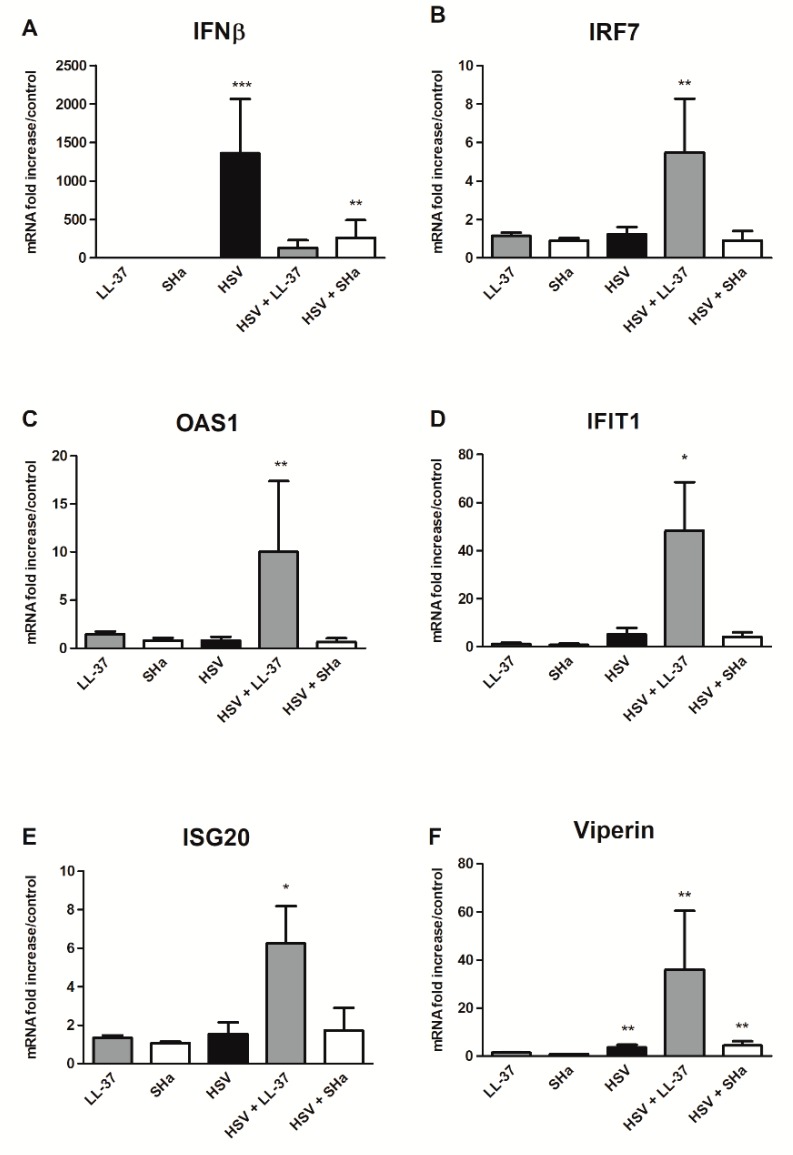
Modulation of the HSV-induced anti-viral response of human primary keratinocytes by LL-37 (2.5 µM) and SHa (10 µM). IFNβ (**A**), IRF7 (**B**), OAS1 (**C**), IFIT1 (**D**), ISG20 (**E**) and viperin (**F**) mRNA expression was quantified in human primary keratinocytes incubated with AMPs alone and 24 h post-infection with HSV-1 at an MOI of 0.1 in presence or absence of AMPs. Data are represented as mean + SEM of at least three independent experiments. **p* < 0.05, ***p* < 0.01 and ****p* < 0.01 compared with the uninfected control.

**Figure 4 viruses-11-00077-f004:**
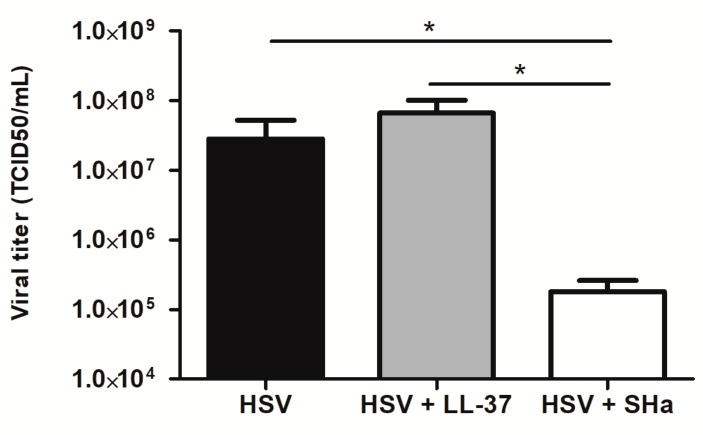
Evaluation of virucidal properties of LL-37 and SHa on HSV-1. HSV-1 suspension was pre-incubated in absence or presence of LL-37 (2.5 µM) and SHa (10 µM) for 1 h at 37 °C before titration by end-point dilution assay on Vero cells. Viral titers are expressed in TCID_50_/mL. Data are represented as mean + SEM of three independent experiments. **p* < 0.05.

**Table 1 viruses-11-00077-t001:** Sequences of the primers used for transcriptomic analysis of the innate anti-viral immune response in keratinocytes.

**IFNB1**	Interferon beta 1	Forward	ATT GCT CTC CTG TTG TGC TCT CC
		Reverse	TGC GGC GTC CTC CTT CTG G
**IRF7**	Interferon regulatory factor 7	Forward	TAC CAT CTA CCT GGG CTT CG
		Reverse	GCT CCA TAA GGA AGC ACT CG
**IFIT1**	Interferon-induced protein with tetratricopeptide repeats 1	Forward	AGT CGT AGA AAG AAC AAT GCA AGA C
		Reverse	TCA TTC ATA ATT TCC TTC CAA TTT GT
**RSAD2**	Radical S-adenosyl methionine domain containing 2	Forward	GGC AAG TTG GTG AGG TTC TG
		Reverse	ACC ACC TCC TCA GCT TTT GA
**OAS 1**	2′-5′-oligoadenylate synthetase 1	Forward	TTG ACT GGC GGC TAT AAA CC
		Reverse	GAG CTC CAG GGC ATA CTG AG
**ISG20**	Interferon-stimulated gene 20 kDa protein	Forward	TGA GGG AGA GAT CAC CGA TT
		Reverse	TAG CCG CTC ATG TCC TCT TT

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
