# Peer review of "Comparison of Anti-Viral Activity of Frog Skin Anti-Microbial Peptides Temporin-Sha and [K3]SHa to LL-37 and Temporin-Tb against Herpes Simplex Virus Type 1"

_viruses, 2019, doi:10.3390/v11010077_

Round 1
Reviewer 1 Report
The present study aimed to evaluate the anti-HSV1 activity of antimicrobial peptides from skin of frog. The issue appears interesting. The introduction well describes both background and aims of the study. The methods are well conducted and fully described. Results are well described but not exhaustively discussed. In particular, despite an extensive discussion of antiviral activity of LL-37, the results concerning the anti-HSV activity of SHa and [K3]SHa are not well described. More justifications for the observed results should be provided. Also, what is the activity on drug-resistant virus? If no further experiments have been performed or if evidences are available, it should be indicated.
Moreover, as a minor issues:
when more than 2 consecutive references are given in the same sentence they should be numbered as follow: [1-4] and not [1,2,3,4]. See lines #78,80,351,356,363,368
Project identification code of the approved Ethics Committee protocol should be stated
Lines #333-345 should be moved from discussion to introduction section.
In this referee's opinion, the present manuscript with suggested revisions, is available for publication on Viruses Journal.
Author Response
Reviewer 1
The present study aimed to evaluate the anti-HSV1 activity of antimicrobial peptides from skin of frog. The issue appears interesting. The introduction well describes both background and aims of the study. The methods are well conducted and fully described. Results are well described but not exhaustively discussed. In particular, despite an extensive discussion of antiviral activity of LL-37, the results concerning the anti-HSV activity of SHa and [K3]SHa are not well described. More justifications for the observed results should be provided. Also, what is the activity on drug-resistant virus? If no further experiments have been performed or if evidences are available, it should be indicated.
A more exhaustive discussion of the results obtained with SHa and [K3]SHa was added to the study lines 388-401. We describe in a more detailled way why we think that the temporin-SHa acts mainly during the pre-fusion step of the virus replication cycle, directly on the virus particle. This conclusion is reinforced by the additional experiments carried out following the recommendations of the reviewer 2 showing that the addition of SHa, 1 hour after the infection of keratinocytes by HSV-1, has no impact on the viral replication.
We did not test drug-resistant viruses. However, since current anti-HSV-1 drugs are nucleoside analogs targeting the viral DNA polymerase, we can assume that the mechanism of action of SHa, virucidal and extracellular, rules out the possibility of cross-resistance between these different antivirals. This point was also added to the discussion lines 414-418.
Moreover, as a minor issues:
when more than 2 consecutive references are given in the same sentence they should be numbered as follow: [1-4] and not [1,2,3,4]. See lines #78,80,351,356,363,368
This has been done.
Project identification code of the approved Ethics Committee protocol should be stated.
Project identification code of the approved Ethics Committee protocol was added to the text (CPP Ouest III - DC-2014-2109).
Lines #333-345 should be moved from discussion to introduction section.
Lines #333-345 have been moved from discussion to introduction.
In this referee's opinion, the present manuscript with suggested revisions, is available for publication on Viruses Journal.
Reviewer 2 Report
This manuscript by Roy et al., describes the effect of anti-microbial peptides (AMPs) on the replication of Herpes Simplex Virus 1 (HSV-1). The use of AMPs is a promising way to treat viral infections for which molecules are often lacking. Although this is not the case for Herpesviruses, for which efficient molecules exist, the development of alternative treatments is necessary to overcome resistances. In this context, this study is relevant to the virology field. The manuscript is well written and the few experiments presented appear to be well executed. The use of primary keratinocytes is an asset, especially for an infection which could be expected to be treated locally.
However, although the scope of the study is promising, I feel that the quantity of data is too limited for the manuscript to warrant publication as an full article and it should be completed with a few additional experiments.
Here are suggestions as to how to improve the manuscript:
Major points:
1) Throughout the paper, the authors use different timings for incubation of the AMPs with the cells, which does not help with the general interpretation. Starting with Figure 2, cells are incubated with the AMPs 1h before infection. The effects on replication seen here are modest (2-3 fold). In Figure 4, the SHa peptide seems to have a much more important effect on virus infectivity. How do the authors explain the 2-3 fold intracellular effect seen on viral replication in Figure 2 which is supposedly independent from the one described in Figure 4?
2) In line with this, an important experiment to include to strengthen this study would be the follow-up of the effects of the molecules on different stages of viral replication by immuno-fluorescence. Cells could be treated 1h before infection with the different AMPs, or 1h after adsorption, or pre-incubated with the virus (at a high moi, such as 10 pfu/ml). Then viral replication could be followed at 6h, 12h and 24h pi using antibodies directed against an immediate-early protein (such as ICP4 or ICP0) and a late protein (such as VP5). This would establish the percentage of infected cells while at the same time help to ascertain where and when each peptide seems to impact viral replication. For instance, SHa could have a moderate effect on entry (saturation of receptors, intracellular transport, etc…) that would show up at 6h pi with less infected cells (ICP4+) and viral particles accumulating at the plasma membrane (VP5+), or in the cytoplasm, or at the nucleus, etc…
3) In general, the paper lacks for basic virology tests to better characterize the antiviral effects of their peptides. In addition to the experiment describe in point #3, they should add at least one basic test such as the effect of their peptides on single or multiple-steps growth curves or on viral plaque size. If these cannot be performed on primary keratinocytes, perhaps the authors could consider trying on a keratinocyte-derived cell line such as HaCat cells which are routinely used in herpesvirology for this kind of purpose.
4) The modified version of SHa, named K3-SHa, is barely described here. The authors do not include this molecule in their characterization assays described in Fig.3 and 4. This is a pity since the molecule seems to be as active as SHa (that is, modestly active on viral replication as described in Fig2) at a lower and more useful concentration than SHa (Fig.2). Moreover, the authors do include this molecule in their title, abstract (l.39) and introduction as if being described as SHa is while this is not the case.
5) Figure 3 should include a control showing the effect of LL-37 alone on the induction of antiviral response.
Minor points :
1) Figure 4 opens the so-far limited results to a larger prospect with the potential of SHa (and K3?) to act directly on the viral particle. Sadly, the paper stops here. Why not include some EM data showing how the viral particles look like after incubation with SHa and LL-37?
2) The whole first paragraph of the discussion would be more appropriate in the introduction. Moreover, the whole point of using primary keratinocytes for this study is justified as late as in lines 346-347. I feel that it should be stated in the first description of results using these cells, so that the reader immediately understands the strength of the results described afterwards.
3) I am puzzled by the statistics used here. First, presentation of the data as means +/- SEM is not as accurate as means +/- SD (standard deviation). More importantly, the statistic tests used as described in the Methods are confusing. Why use a one-way ANOVA test (a parametric test used essentially for comparison of multiple groups of sample) for comparing only two groups and make a follow-up with two non-parametric tests? Either the data have a normal (Gaussian) distribution, in which case a parametric test for two groups is required (such as Student’s t test) or the data does not have a normal distribution and a non-parametric test for two groups is appropriate (such as Mann-Withney). Please clarify.
4) The scale used in Figure 2 does not make it easy to estimate the extent of the effect of LL-37. In addition, numbers and percentages should be stated in the results rather than vague estimations such as “causing a decrease of about one log”.
5) The use of end-point dilutions to estimate the titer of HSV-1 is unusual since the virus plaques very well in normal cell culture conditions, in particular in Vero cells. Could the authors explain the choice of TCID50 to estimate the titer of their stocks?
6) The authors use a clinical strain of HSV-1, which is perfectly relevant for this study using primary keratinocytes. I was therefore surprised to see that virus stocks were produced on Vero cells, a cell line where HSV-1 is strongly cell-associated and that facilitates strong selection for cell-culture adapted viruses. The use of Vero to obtain cell-free virus could explain the relatively low viral titers obtained in this study. The MRC-5 cell line on which the virus was originally isolated would have been more appropriate. Please justify.
Some typos:
- l.66: the authors probably mean “b-pleated sheets”
- l.114: “cellular debris WERE removed”
- l.160 : “The antiviral properties were evaluated by evaluating”, probably needs rephrasing!
- in general: cells are inoculated with an moi of 0.1 TCID50/mL rather than with 0.1 moi
- l.245: “viral DNA concentration per ng OF total cell nucleic acids”.
- l.397: “genomewide” for “genome-wide”
- l.402: “solve the problem of Route of administration”

Author Response
Reviewer 2
This manuscript by Roy et al., describes the effect of anti-microbial peptides (AMPs) on the replication of Herpes Simplex Virus 1 (HSV-1). The use of AMPs is a promising way to treat viral infections for which molecules are often lacking. Although this is not the case for Herpesviruses, for which efficient molecules exist, the development of alternative treatments is necessary to overcome resistances. In this context, this study is relevant to the virology field. The manuscript is well written and the few experiments presented appear to be well executed. The use of primary keratinocytes is an asset, especially for an infection which could be expected to be treated locally.
However, although the scope of the study is promising, I feel that the quantity of data is too limited for the manuscript to warrant publication as an full article and it should be completed with a few additional experiments.
Here are suggestions as to how to improve the manuscript:
Major points:
1) Throughout the paper, the authors use different timings for incubation of the AMPs with the cells, which does not help with the general interpretation. Starting with Figure 2, cells are incubated with the AMPs 1h before infection. The effects on replication seen here are modest (2-3 fold). In Figure 4, the SHa peptide seems to have a much more important effect on virus infectivity. How do the authors explain the 2-3 fold intracellular effect seen on viral replication in Figure 2 which is supposedly independent from the one described in Figure 4?
The direct extracellular antiviral activity assessed in Figure 4 was observed to be far superior to the reduction of the HSV-1 replication measured at 24h post-infection of keratinocytes treated with SHa 1 h prior infection as depicted in Figure 2 (2 logs vs. two-fold reduction as indicated by the reviewer). In the second assay (Figure 2), both the extra- and intracellular antiviral effects of the peptide were investigated with a specific focus on the antiviral effects related to modifications of the target cell (saturation of receptors, stimulation of the innate immune response…). In this assay, the virus particle was less exposed to the direct inactivation of its infectivity than when pre-incubated with the peptide, likely because of a rapid intracellular virus penetration into keratinocyte following inoculation. This could explain the lower antiviral activity of SHa observed in Figure 2 than in Figure 4 in connection with its extracellular virucidal activity. Conversely, the higher antiviral activity of LL-37 observed in Figure 2 than in Figure 4 can be linked to its antiviral mechanism of action through modulation of the cell innate immune response. This point was added to the discussion lines 391-401.
2) In line with this, an important experiment to include to strengthen this study would be the follow-up of the effects of the molecules on different stages of viral replication by immuno-fluorescence. Cells could be treated 1h before infection with the different AMPs, or 1h after adsorption, or pre-incubated with the virus (at a high moi, such as 10 pfu/ml). Then viral replication could be followed at 6h, 12h and 24h pi using antibodies directed against an immediate-early protein (such as ICP4 or ICP0) and a late protein (such as VP5). This would establish the percentage of infected cells while at the same time help to ascertain where and when each peptide seems to impact viral replication. For instance, SHa could have a moderate effect on entry (saturation of receptors, intracellular transport, etc…) that would show up at 6h pi with less infected cells (ICP4+) and viral particles accumulating at the plasma membrane (VP5+), or in the cytoplasm, or at the nucleus, etc…
In order to unravel the effects of SHa and LL-37 on different stages of viral replication and, in particular, on post-fusion steps, we performed additional experiments in which AMPs were added one hour after the inoculation of HSV-1 on human keratinocytes. These experiments performed, as the others, in three biological replicates using, each, primary keratinocytes from different individuals showed that adding the two peptides 1 h post-inoculation did not modify the viral DNA concentrations in cell monolayer after 24 h of incubation by comparison to untreated infected cells. These results showed the absence of effect of SHa on the post-fusion steps of the HSV-1 replication cycle supporting an antiviral activity primarily during the pre-fusion steps. This was added to the discussion lines 399-401 as data not shown. They also suggest that the antiviral activity of LL-37, by immunomodulation, only occurs when the peptide is added prior to infection, on uninfected cells. This second point is worth checking in a second set of experiments and was therefore not discussed further in the present study.
In the same way, we also conducted a transcriptomic analysis of immediate early (UL54: ICP27), early (UL23: Thymidine Kinase) and late (UL19: VP5) viral genes at 6 and 24 h in keratinocytes treated with LL-37 and SHa 1 h before and 1 h after HSV-1 inoculation. We observed that virus transcription was significantly reduced at 6 and 24 h in keratinocytes treated with LL-37 prior to HSV-1 infection in good correlation with the reduction of viral DNA replication assessed in this condition. On the other hand, virus transcription was not altered in keratinocytes pre-treated with SHa as well as in keratinocytes treated 1 h post-inoculation with each of the two peptides. Again, these very preliminary results, however obtained from three independent biological replicates, led us not to include them in this article.
3) In general, the paper lacks for basic virology tests to better characterize the antiviral effects of their peptides. In addition to the experiment describe in point #3, they should add at least one basic test such as the effect of their peptides on single or multiple-steps growth curves or on viral plaque size. If these cannot be performed on primary keratinocytes, perhaps the authors could consider trying on a keratinocyte-derived cell line such as HaCat cells which are routinely used in herpesvirology for this kind of purpose.
Overall, antiviral properties of LL-37 and SHa against HSV-1 were tested by addition one hour before or one hour after infection of human primary keratinocytes, and by measurement of virucidal activity after pre-incubation of the virus with the peptides before titration on Vero cells. In addition, we assessed their immunomodulatory properties during HSV-1 keratinocyte infection. We showed, for the first time, anti-HSV-1 activity of temporins-SH, mainly extracellular and virucidal, and demonstrated that LL-37 can fight HSV-1 infection through induction of cell antiviral response. These experiments are indeed worthy of further investigation, including EM and immunofluorescence assays for monitoring the translation of viral antigens, but whose implementation is incompatible with the time allowed for the revision of this manuscript.
4) The modified version of SHa, named K3-SHa, is barely described here. The authors do not include this molecule in their characterization assays described in Fig.3 and 4. This is a pity since the molecule seems to be as active as SHa (that is, modestly active on viral replication as described in Fig2) at a lower and more useful concentration than SHa (Fig.2). Moreover, the authors do include this molecule in their title, abstract (l.39) and introduction as if being described as SHa is while this is not the case.
[K3]SHa was only used in the figures 1 and 2 because of higher cytotoxicity and similar antiviral activity to SHa at the maximum non cytotoxic dose. Moreover, [K3]SHa sequence is almost identical to SHa with only one amino-acid substitution in order to increase the net positive charge of SHa to enhance electrostatic interaction with the negatively charged membrane of the microorganisms. It is therefore very likely that its mode of action is identical to that of SHa. For these reasons, we focused our work on the temporin-SHa and not on [K3]SHa, which was, however, tested here for the first time against a virus and thus deserves to appear also in the title of this study.
5) Figure 3 should include a control showing the effect of LL-37 alone on the induction of antiviral response.
RNA expression assessed in uninfected cells treated with LL-37 or SHa alone was added to figure 3 in order to depict pro-inflammatory properties of the peptides themselves.
Minor points :
1) Figure 4 opens the so-far limited results to a larger prospect with the potential of SHa (and K3?) to act directly on the viral particle. Sadly, the paper stops here. Why not include some EM data showing how the viral particles look like after incubation with SHa and LL-37?
Observation using electron microscopy were performed but were not conclusive enough to be included in this study. We will need more time to acquire the necessary know-how in electron microscopy to generate images of sufficient quality.
2) The whole first paragraph of the discussion would be more appropriate in the introduction. Moreover, the whole point of using primary keratinocytes for this study is justified as late as in lines 346-347. I feel that it should be stated in the first description of results using these cells, so that the reader immediately understands the strength of the results described afterwards.
As also suggested by the reviewer 1, lines #333-345 have been moved from discussion to introduction. Moreover, a description of keratinocytes as HSV-1 target cells in cutaneous and mucosal lesions has been added at the end of the first chapter of the introduction lines 55-56.
3) I am puzzled by the statistics used here. First, presentation of the data as means +/- SEM is not as accurate as means +/- SD (standard deviation). More importantly, the statistic tests used as described in the Methods are confusing. Why use a one-way ANOVA test (a parametric test used essentially for comparison of multiple groups of sample) for comparing only two groups and make a follow-up with two non-parametric tests? Either the data have a normal (Gaussian) distribution, in which case a parametric test for two groups is required (such as Student’s t test) or the data does not have a normal distribution and a non-parametric test for two groups is appropriate (such as Mann-Withney). Please clarify.
In the previous version of the manuscript, a Kurskall-Wallis test followed by the Dunn’s test (a non-parametric test which appears in the one-way ANOVA section of GraphPad Prism) has been used for cytotoxicity evaluation (Figure 1) whereas other experiments have been analyzed using Mann-Whitney tests (Figure 2 to 4). In order to clarify this point, all the statistical analyses have been redone using the Mann-Whitney test (from Figure 1 to Figure 4). The statistic chapter in the Materials and Methods section has been corrected accordingly (page 6, lines 222-225).
4) The scale used in Figure 2 does not make it easy to estimate the extent of the effect of LL-37. In addition, numbers and percentages should be stated in the results rather than vague estimations such as “causing a decrease of about one log”.
The scale used was that allowing the best visualization of the antiviral effects observed (contrary to the logarithmic scale). The viral load values as well as the percentages of reduction with and without AMP treatment have been added to the text as requested.
5) The use of end-point dilutions to estimate the titer of HSV-1 is unusual since the virus plaques very well in normal cell culture conditions, in particular in Vero cells. Could the authors explain the choice of TCID50 to estimate the titer of their stocks?
The titer of the virus stock was determined by end-point dilution assay because it is the titration method routinely used in our laboratory. This method is not the most widely used in the literature for the determination of HSV-1 titer, but a few studies have also used it.
6) The authors use a clinical strain of HSV-1, which is perfectly relevant for this study using primary keratinocytes. I was therefore surprised to see that virus stocks were produced on Vero cells, a cell line where HSV-1 is strongly cell-associated and that facilitates strong selection for cell-culture adapted viruses. The use of Vero to obtain cell-free virus could explain the relatively low viral titers obtained in this study. The MRC-5 cell line on which the virus was originally isolated would have been more appropriate. Please justify.
This is a suggestion we will follow in the future since the clinical strain was also isolated on MRC5 cells. However, we performed only two passages of the virus strain on Vero cells, one for isolation and one for amplification. Moreover, the virus titer obtained using Vero cell was high enough to allow keratinocyte infection at an MOI of 0.1 TCID50 per cell. This is the reason why we did not look at a more efficient cell culture model for virus stock production.
Some typos:
- l.66: the authors probably mean “b-pleated sheets”
This has been corrected.
- l.114: “cellular debris WERE removed”
This has been corrected.
- l.160 : “The antiviral properties were evaluated by evaluating”, probably needs rephrasing!
The sentence has been rephrased.
- in general: cells are inoculated with an moi of 0.1 TCID50/mL rather than with 0.1 moi
We precised the first time that cells were inoculated with an MOI of 0.1 TCID50 per cell and then indicated 0.1 MOI.
- l.245: “viral DNA concentration per ng OF total cell nucleic acids”.
This has been corrected.
- l.397: “genomewide” for “genome-wide”
This has been corrected.
- l.402: “solve the problem of Route of administration”
This has been corrected.

Reviewer 3 Report
The manuscript by Roy et al describing the properties of temporin-SHa and [K3]SHa against herpes simplex virus demonstrate an antiviral function of both variants of SHa. The manuscript is generally well written.
Minor comments
p 3, line 128 - Are the Keratinocytes used in this study from different individuals in each biological replicate or are the same cells used?
Figure 3 - Are the uninfected control cells treated with AMPs? The data in this figure would be more clear if the RNA expression would be demonstrated in HSV-infected cells with and without AMP as well as in uninfected cells treated with and without AMP.
p10, line 369 - induced in keratinocytes infected with LL-37
Author Response
Reviewer 3
The manuscript by Roy et al describing the properties of temporin-SHa and [K3]SHa against herpes simplex virus demonstrate an antiviral function of both variants of SHa. The manuscript is generally well written.
Minor comments
p 3, line 128 - Are the Keratinocytes used in this study from different individuals in each biological replicate or are the same cells used?
Yes, primary keratinocytes used in each biological replicate were from different individuals. This has been indicated page 4 lines 165-166.
Figure 3 - Are the uninfected control cells treated with AMPs? The data in this figure would be more clear if the RNA expression would be demonstrated in HSV-infected cells with and without AMP as well as in uninfected cells treated with and without AMP.
As also requested by reviewer 2, RNA expression assessed in uninfected cells treated with AMPs alone was added to figure 3 in order to depict pro-inflammatory properties of the peptides themselves.
p10, line 369 - induced in keratinocytes infected with LL-37
This has been corrected.
Round 2
Reviewer 2 Report
To my opinion, the main weakness of the original manuscript was that it lacked basic virology tests to ascertain the promising effects of the AMPs as described here. In the revised form of the manuscript, the authors did not provide additional data such as suggested in my major points #2 or #3, yet those assays are very straightforward and not time-consuming. For instance, to determine the size of viral plaques in presence or absence of AMPs would indicate a general effect of the AMPs on viral replication and spread, which is relevant in a clinical-orientated article such as this one. If the target of this study is to show that the AMPs presented here are good candidates for antiviral therapy, what is the importance of a two-fold effect on viral replication (Figure 2) if this effect does not translate into a significant biological impact such as an impaired viral spread?
The authors point for the lack of time available to carry out some of these assays. However, establishing virus plaques in presence or absence of the AMPs takes 3 days and quantifying the plaque size two more days, so it is unclear to me why the authors refuse to carry out at least one of the assays suggested.
All in all, the whole paper’s promises on the antiviral activities of these AMPs (according to the title and abstract and conclusions) are mostly based on a single figure (Figure 2 since [K3SHa] is included in the title, abstract and conclusions and not included in Figure 4). The results in this figure are a promising start but certainly not enough to be the core of a whole paper.
Therefore, I still feel that the manuscript lacks the depth of a complete investigation and is more a preliminary report (or a short note) rather than a complete research article.
Author Response
We agree with the reviewer to consider this study as a preliminary work and submit it as a short communication rather than a full-length paper.